# Development of Fluorescent Reagent Based on Ligand Exchange Reaction for the Highly Sensitive and Selective Detection of Dopamine in the Serum

**DOI:** 10.3390/s19183928

**Published:** 2019-09-12

**Authors:** Yoshio Suzuki

**Affiliations:** Health Research Institute, National Institute of Advanced Industrial Science and Technology (AIST), 1-1-1 Higashi, Tsukuba, Ibaraki 305-8566, Japan; suzuki-yoshio@aist.go.jp; Tel.: +81-29-861-6122

**Keywords:** fluorescence, dopamine, sensor, molecular probe, serum, high-throughput analysis

## Abstract

A new fluorescent probe (**BDP-Fe^2+^**) was developed for targeting dopamine, with a boron–dipyrromethenyl (BDP) group as the fluorophore and a Fe^2+^ complex as the ligand exchange site. The free form of **BDP-Fe^2+^** in solution displayed weak fluorescence emission, while it showed strong fluorescence emission after interaction with dopamine due to the release of Fe^2+^ from **BDP-Fe^2+^**, confirming the binding of Fe^2+^ to dopamine. The increase in fluorescence intensity was concentration-dependent, and a good linear relationship was observed between the fluorescence intensity and dopamine concentration. The detection limit of dopamine by **BDP-Fe^2+^** was 1.1 nM, indicating a 20-fold higher sensitivity than that of previously reported compounds. The reaction of **BDP-Fe^2+^** with dopamine was not affected by the presence of foreign substances, allowing the highly selective detection of dopamine in the human serum sample. The results of this study indicate that the novel compound **BDP-Fe^2+^** is a reliable fluorescent molecular probe for the detection of dopamine and can be widely employed in diverse scientific areas.

## 1. Introduction

Dopamine is an organic compound of the catecholamine and phenylethylamine families. Dopamine plays an important role as a neurotransmitter, and it is responsible for a wide range of cognitive functions, including motivation, behavior, learning, and memory. Its concentration affects emotions and reflects the state of one’s health [1,2,3]. It greatly influences metabolism and plays a key role in maintaining central nervous system functions, including neuroplasticity, attention span, and memory, as well as various cardiovascular, hormonal, and renal functions [4]. Although dopamine itself is not a toxicant, it causes many diseases, including various diseases related to the human brain and neurological disorders such as Parkinson’s disease; Parkinson’s disease causes a reduction of dopaminergic neurons, and insufficient dopamine is then made in the human body. As a result, the movement of the body cannot be adjusted and is impaired; this disease increasingly affects the general population and is attracting considerable attention [5,6,7,8]. Consequently, there is an urgent need to develop efficient, rapid, sensitive, and selective methods for detecting dopamine and enabling the rapid development of related medication used in nerve physiology.

Various studies have been performed on the optical determination of dopamine from the viewpoint of the native fluorescence properties of dopamine, the efficiency of fluorescence quenching based on the reaction between dopamine and HRP (horseradish peroxidase), the utilization of quantum dots, and the derivatization of dopamine by ethylenediamine following oxidation using mercury(II) nitrate [9,10,11,12,13,14].

However, these methods are neither highly selective nor highly sensitive; instead, they are time-consuming and require further separation processes such as ion exchange chromatography, thin-layer chromatography, or high-performance liquid chromatography.

The prerequisites considered in the present study to design a fluorescent reagent for dopamine detection were as follows:The generation of a strong fluorescence signal after reaction with the target molecule;The elimination of background noise for the highly sensitive detection of dopamine;The reduction of interference from foreign substances for highly selective dopamine assay. 

We have previously developed fluorometric reagents for the detection of various biological substances containing dopamine [15,16,17,18,19,20,21]. The interaction between the fluorescent reagent and dopamine caused the enhancement of fluorescence intensity due to the release of Fe^2+^ from the reagent and subsequent formation of a complex between Fe^2+^ and dopamine. However, the analysis using this fluorescent reagent had poor sensitivity and low photostability due to the chemical properties of the cyanopyranyl moiety, which could not be applicable in long-term analysis.

In this study, a new fluorescent molecular probe has been developed with the boron–dipyrromethenyl (BDP) group as the fluorescent emitter [22,23] and an imino-di-acetic acid-Fe^2+^ complex both as a fluorescent quencher and a ligand exchange moiety for the detection of dopamine. The BDP group has interesting spectral properties such as high quantum yields, large extinction coefficients, and high photostability—characteristics that satisfy the above concept of molecular design [24,25]. The chemical structures of the fluorescent reagents (BDP and its Fe^2+^ complex, BDP-Fe^2+^) are shown in Figure 1. The interactions of these molecules with dopamine were investigated, along with their selectivity, sensitivity, photostability, and detection in human serum samples. The results clearly indicated that BDP-Fe^2+^ acts as a reliable fluorescent probe, and it is useful for the high-throughput analysis of dopamine.

## 2. Materials and Methods

### 2.1. Chemicals, Materials, and Apparatus

All the chemicals required for the synthesis and fluorometric measurements were of analytical grade and provided from Tokyo Chemical Industry (TCI, Tokyo, Japan), Wako Pure Chemical Industries, Ltd. (Osaka, Japan) and Sigma Aldrich (St. Louis, MO, USA). Absorption spectra were recorded at 25 °C on a V-670 UV/Visible Spectrophotometer (JASCO, Tokyo, Japan). Fluorescence spectra were recorded at 25 °C on a JASCO FP-6500 fluorophotometer and an FMP-825 microplate reader. ^1^H nuclear magnetic resonance (NMR) spectra were recorded on a Bruker AV-500 spectrometer.

### 2.2. Measurements

The compound BDP-Fe^2+^ was dissolved in 4-(2-hydroxyethyl)-1-piperazineethanesulfonic acid (HEPES) buffer solution (pH 7.0) to obtain a final concentration of 1.0 µM. After the addition of 1.0 mL of dopamine or foreign substance solution to 1.0 mL of BDP-Fe^2+^ solution, the reaction mixture was incubated for 5 min, and fluorescence spectra were recorded.

### 2.3. Synthesis of the Ligand

The synthesis of the fluorescent reagent is illustrated in Scheme 1. The protocols for the synthesis of diethyl-2,2’-(5-formyl-2-hydroxybenzylazanediyl)diacetate have been reported previously [19]. The individual synthetic protocols are described below.

#### 2.3.1. Diethyl 2,2’-((5-((3,5-dimethyl-1H-pyrrol-2-yl)(3,5-dimethyl-2H-pyrrol-2-ylidene)methyl)-2-hydroxybenzyl)azanediyl)(Z)-diacetate

Anhydrous dichloromethane (40 mL) and 2,4-dimethyl pyrrole (2.5 g, 26.4 mmol) were added to a solution of diethyl-2,2’-(5-formyl-2-hydroxybenzylazanediyl)-diacetate (2.2 g, 13.2 mmol) and stirred for 30 min in an N_2_ atmosphere. After the addition of trifluoroacetic acid (0.5 mL, 6.5 mmol), the reaction mixture was stirred for 24 h at room temperature. 2,3-Dichloro-5,6-dicyano-1,4-benzoquinone (5.97 g, 26.26 mmol), dissolved in a mixture of anhydrous THF (25 mL) and anhydrous dichloromethane (25 mL), was added dropwise over 15 min to the mixture, which was then stirred for 4 h at room temperature. Saturated aqueous NaHCO_3_ solution was added and the mixture was extracted with dichloromethane. The organic layer was washed with water and brine, dried over Na_2_SO_4_, and concentrated in vacuo. The crude product was purified by column chromatography (SiO_2_, *n*-hexane:ethyl acetate:triethylamine = 49:49:2 *v*/*v*) to obtain a brown solid with 27.5% yield.

^1^H NMR (CDCl_3_, 500MHz, r.t., TMS, *δ*/ppm) 1.24 (6H, t), 1.27(6H, s), 2.34(6H, s), 3.53 (4H, s), 4.04 (2H, s), 4.21 (4H, q), 5.88(2H, s), 6.99 (1H, d), 7.57 (1H, s), 7.75 (1H, d), 10.54 (1H, bs).

#### 2.3.2. Diethyl 2,2’-((5-(5,5-difluoro-1,3,7,9-tetramethyl-5H-4λ4,5λ4-dipyrrolo[1,2-c:2’,1’-f] [1,3,2] diazaborinin-10-yl)-2-hydroxybenzyl) azanediyl) diacetate

Diethyl 2,2’-((5-((3,5-dimethyl-1H-pyrrol-2-yl) (3,5-dimethyl-2H-pyrrol-2-ylidene)methyl)-2-hydroxybenzyl) azanediyl) (Z)-diacetate (1.6 g, 4.78 mmol) was dissolved in anhydrous toluene (65 mL), triethylamine (9.2 mL), and BF_3_•Et_2_O (9.2 mL). The mixture was stirred for 3 h at room temperature. Saturated aqueous NaHCO_3_ solution was added and the mixture extracted with CH_2_Cl_2_. The organic layer was washed with brine, dried over Na_2_SO_4_, and concentrated in vacuo. The crude product was purified by column chromatography (SiO_2_, *n*-hexane:ethyl acetate = 1:1 *v*/*v*) to obtain a brown solid in 95.8% yield.

^1^H NMR (CDCl_3_, 500 MHz, r.t., TMS, δ/ppm) 1.25 (6H, t), 1.36(6H, s), 2.55(6H, s), 3.54 (4H, s), 4.03 (2H, s), 4.22 (4H, q), 5.95 (2H, s), 6.98 (1H, d), 7.58 (1H, s), 7.76 (1H, d), 10.55 (1H, bs).

#### 2.3.3. 2,2’-((5-(5,5-Difluoro-1,3,7,9-tetramethyl-5H-4λ4,5λ4-dipyrrolo[1,2-c:2’,1’-f][1,3,2]diazaborinin-10-yl)-2-hydroxybenzyl)azanediyl) diacetic acid (BDP)

A mixture of diethyl 2,2’-((5-(5,5-difluoro-1,3,7,9-tetramethyl-5H-4λ4,5λ4-dipyrrolo[1,2-c:2’,1’-f][1,3,2], diazaborinin-10-yl)-2-hydroxybenzyl) azanediyl) diacetate (0.5 g, 1.31 mmol), and sodium hydroxide (0.30 g, 2.17 mmol) in EtOH–water (40 mL–10 mL) was stirred for 24 h at room temperature. After the removal of EtOH, the solution was acidified to pH 6.0 using 1 N HCl, extracted with ethyl acetate, and dried over Na_2_SO_4_. The solvent was evaporated in vacuo. The brown solid obtained in 96.2% yield was used without further purification.

^1^H NMR (CD_3_OD, 500 MHz, r.t., TMS, δ/ppm) 1.37 (6H, s), 2.56 (6H, s), 3.55 (4H, s), 4.04 (2H, s), 5.96 (2H, s), 6.99 (1H, d), 7.57 (1H, s), 7.75 (1H, d), 10.54 (1H, bs).

#### 2.3.4. 2,2’-((5-(5,5-Difluoro-1,3,7,9-tetramethyl-5H-4λ4,5λ4-dipyrrolo[1,2-c:2’,1’-f][1,3,2]diazaborinin-10-yl)-2-hydroxybenzyl)azanediyl)diacetic acid Fe(II) complex (BDP-Fe^2+^)

To a solution of 2,2’-((5-(5,5-difluoro-1,3,7,9-tetramethyl-5H-4λ4,5λ4-dipyrrolo[1,2-c:2’,1’-f][1,3,2]diazaborinin-10-yl)-2-hydroxybenzyl)azanediyl)diacetic acid (84.1 mg, 0.13 mmol) in methanol (20 mL), 100 mM FeCl_2_ aqueous solution (1.3 mL) was added, and the mixture was stirred for 1 h at room temperature. After solvent removal, the residue was dried, and the product was recovered as a brown solid in 98% yield.
ESI-MS (+): [M]^2+^ = 271.05

## 3. Results and Discussion

The excitation and emission spectra of BDP were recorded in a buffer solution (pH 7.0) at 25 °C to study the photophysical properties in vitro. Typical excitation and emission spectra of BDP are shown in Figure 2, with excitation and emission maxima at 496 nm and 525 nm, respectively. This result was consistent with the typical excitation and emission maxima of a BDP fluorophore.

To study the reaction of Fe^2+^ with BDP, fluorescence spectra were investigated, and the signal was monitored by mixing BDP (1.0 µM) with Fe^2+^ (5.0 µM) in HEPES (20.0 mM; pH 7.2) to form BDP-Fe^2+^. The fluorescence intensities of BDP decreased following the addition of Fe^2+^ due to the considerable quenching effect, as shown in Figure 3. Other metal cations (Ni^2+^, Co^2+^, Cu^2+^, Mn^2+^, Zn^2+^, Li^+^, Na^+^, K^+^, Cs^+^, Mg^2+^, Ca^2+^, Ba^2+^, and Pb^2+^) were added to the solution of BDP, and the fluorescence intensities were monitored. The observed order of the fluorescence intensity was as follows: Fe^2+^ (6.5) > Ni^2+^ (25.0) > Co^2+^ (156.0) > Cu^2+^(201.5) > Mn^2+^ (250.2) > Zn^2+^ (292.3). Fluorescence quenching did not take place for alkali metal ions, alkaline earth metal ions, and Pb^2+^, as shown in Figure 3b. Thus, Fe^2+^ was a particularly strong quencher of BDP fluorescence. To take advantage of this phenomena, the BDP-Fe^2+^ complex was used in this study, because we used the ligand exchange mechanism, and a large fluorescence enhancement from quenching state was requested, enabling the highly sensitive detection of dopamine by the dissociation of Fe^2+^ from BDP-Fe^2+^.

Fluorimetric titrations were performed with dopamine by using BDP-Fe^2+^ to investigate the ligand exchange reaction. The concentration of BDP-Fe^2+^ was 1.0 µM and that of the dopamine solutions varied between 0 and 5.0 µM in HEPES buffer solutions (20.0 mM, pH 7.2). The data are shown in Figure 4. As is evident from the data, the fluorescence enhancement following the addition of dopamine was remarkable. The fluorescence intensity signal was saturated at a dopamine concentration of >4.0 µM, and the numerical values of the fluorescence intensity of BDP-Fe^2+^ increased from 6.4 (free form) at 525 nm to 328.9 nm following the reaction with 5.0 µM of dopamine, corresponding to a 51.4-fold increase in fluorescence intensity. The detection limit was determined to be 1.1 nM of dopamine, which was calculated from the baseline noise (S/N = 3). The detection limit of the previously reported compound is 10.0 nM [19], indicating 10-fold poorer sensitivity than that of BDP-Fe^2+^. These observations indicate that the BDP-Fe^2+^ containing a dipyrromethenyl group shows brighter fluorescence and large fluorescence enhancement after reaction with dopamine, thereby improving sensitivity. Thus, the chemical structures of the fluorophore in BDP-Fe^2+^ play an important role in the detection of dopamine.

Figure 5 shows the reaction time dependence of the fluorescence intensity at 525 nm of BDP-Fe^2+^ with dopamine. The fluorescence intensity saturated after 3.0~5.0 min, and 5.0 min was considered optimal for the reaction between BDP-Fe^2+^ and dopamine.

In order to understand how pH affects the reaction between BDP-Fe^2+^ and dopamine, fluorescence signals were monitored at different pH values. Figure 6 shows the fluorescence intensity at 525 nm of BDP-Fe^2+^ with 5.0 µM of dopamine. At high pH values (10 to 11) and low pH (3 to 4), the reaction of BDP-Fe^2+^ with dopamine was inhibited by the solution pH, whereas the fluorescence intensity was nearly unchanged when the pH was in the range 5–9. As a result, the reaction of BDP-Fe^2+^ with dopamine was neither inhibited nor masked at a physiological level of pH values, and this assay should be appropriate to monitor the quantitative analysis of dopamine under such conditions.

To investigate the role of Fe^2+^ in BDP-Fe^2+^ for the detection of dopamine, the fluorescent properties of the free form of BDP (not Fe^2+^ complex), were monitored before and after the addition of dopamine (0 to 5.0 µM). Fluorescence intensities of BDP at 525 nm hardly changed after the addition of dopamine, as shown in Figure 7. Moreover, other transition metal complexes (BDP-M^2+^ (M = Ni^2+^, Co^2+^, Cu^2+^, Mn^2+^, Zn^2+^)) did not show any response after the addition of dopamine because no reaction occurred between these metals and the dopamine [19]. On the other hand, BDP-Fe^2+^ indicated significant fluorescence enhancement after the reaction with dopamine. From these results, Fe^2+^ in BDP-Fe^2+^ plays an important role in the detection of dopamine, and schematic representations of the reaction between BDP-Fe^2+^ dopamine are illustrated in Figure 8. The free form of BDP-Fe^2+^ emitted remarkably weak fluorescence, whereas the fluorescence enhancement by the interaction of BDP-Fe^2+^ with dopamine can be considered to be the release of Fe^2+^ from BDP-Fe^2+^, and the dopamine coordinated with Fe^2+^ in a 3:1 stoichiometry to form a stable complex by the ligand exchange reaction [9,14].

Figure 9 shows the fluorescence ratio of BDP-Fe^2+^ at 525 nm before and after the addition of dopamine, and other foreign substances that may be present in biological samples. All the tests were carried out with a mixture of BDP-Fe^2+^ and an excess amount of foreign substances. The fluorescence enhancement of BDP-Fe^2+^, when dopamine was added, was considerably larger (I/I_0_ = 49.8) than those after the addition of adrenaline (2.5), noradrenaline (2.2), serotonin (1.3), and other compounds (~1.0). The obtained results indicated that the response of BDP-Fe^2+^ to dopamine was unaffected by the excess amounts of other biological substances due to the release of Fe^2+^ from BDP-Fe^2+^ and formation of a stable Fe^2+^-dopamine complex.

The analytical performance of BDP-Fe^2+^ was compared with those of previously reported compounds (References 1 and 2) and is summarized in Table 1. BDP-Fe^2+^ exhibited large fluorescence enhancement upon dopamine addition, whereas the fluorescence intensities of [1] and [2] were approximately three times and ten times lower, respectively, than those of BDP-Fe^2+^. Thus, BDP-Fe^2+^ could be used to detect dopamine with greater sensitivity than in [1] and [2]. These observations indicate that the BDP group in BDP-Fe^2+^ is a sensitive functional group and contributes to a larger fluorescence enhancement via ligand exchange between dopamine and BDP-Fe^2+^, and the chemical structure of the fluorescent dye in the fluorescent probe plays an important role in determining the fluorescent properties.

The lifetime of BDP-Fe^2+^ is important for the continuous monitoring of dopamine. The BDP-Fe^2+^ solution was irradiated by excitation light (490 nm), and the fluorescence intensities were observed for 24 h. Compared with BDP-Fe^2+^, the photostability of a previously reported compound containing the cyanopyranyl group [19] was observed under the irradiation of excitation light (455 nm) for the same period. The data are shown in Figure 10. After irradiation for 24 h, the fluorescence ratio of BDP-Fe^2+^ was ~98% of the original value, whereas previously reported compounds showed ~64%. This result indicates that BDP-Fe^2+^ is more stable than previously reported compounds, and that this compound produces a reliable quantitative response to dopamine even after long periods of optical measurement.

As an application of this assay, the detection of dopamine in human serum was conducted using BDP-Fe^2+^ and the results were compared with dopamine assays in HEPES buffer solution. Figure 11 shows the relationship between fluorescence intensities at 525 nm and dopamine concentration. The fluorescence intensity of BDP-Fe^2+^ in human serum or in the buffer solution increased with increasing dopamine concentration, and a good linear relationship was observed. Moreover, fluorescence intensities at each dopamine concentration for assay in human serum were almost the same as those in HEPES buffer solution. Thus, BDP-Fe^2+^ could be used to detect dopamine with high selectivity in the presence of other substances as bovine serum albumin (BSA), inorganic salts, etc., and this assay is applicable to the detection of dopamine in the presence of impurities.

This assay was compared with another detection method—i.e., enzyme linked immunosorbent assay (ELISA)—and the results are summarized in Table 2. ELISA is time-consuming, labor-intensive, and has a high background signal/noise ratio and low sensitivity [26]. On the other hand, the fluorescent method described here shows a higher sensitivity, wider detection range, and lower background signal. Therefore, the dopamine analysis using BDP-Fe^2+^ satisfies the requirements for highly sensitive, highly selective, and high-throughput assays of dopamine and can be widely used as a convenient method for dopamine detection in both research laboratories and medical applications.

## 4. Conclusions

In the present study, a new fluorescent sensing molecule that selectively couples with dopamine and exhibits a large increase in fluorescence intensity as a result has been reported. No interference in the detection of dopamine by the presence of foreign substances has been found. In addition, the detection of dopamine in human serum was successfully performed. This assay for dopamine was not only easy to apply, but it has been proved to be rapid, highly sensitive and selective. Based on the extremely positive results of this study, further efforts are ongoing to investigate other possible indicators for the detection of various living substances.

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
