# Peer review of "Development of Fluorescent Reagent Based on Ligand Exchange Reaction for the Highly Sensitive and Selective Detection of Dopamine in the Serum"

_sensors, 2019, doi:10.3390/s19183928_

Round 1

Reviewer 1 Report

The authors developed a new ensemble fluorescent probe based on BDP combined with Fe2+ that exhibits a rapid, sensitive and selective response for dopamine in buffer solution. When dopamine was added the Fe2+ ion was removed from BDP-Fe2+ weak-fluorescent ensemble by the formation of a more stable complex of dopamine with Fe2+ resulting in a fluorescence turn-on of system. The paper is very well structured and written. My recommendation is to be published after minor revisions.

The authors claimed that the probe has good selectivity for dopamine. I wonder that the Fe2+-dopamine complex forms strongly. The authors should discuss it. The interference of other metal ions such as Fe3+, Cu2+, Zn2+ and so on with fluorescence intensity of BDP-Fe2+ should be carefully evaluated. "The quenching efficiency was Fe2+ (6.5) > Ni2+ (25.0) > Co2+ (156.0) shown in Figure 3 (b)" How is the quenching efficiency calculated here? It should be clarified.

Author Response

Answer to Reviewer 1:

Thank you for your specific comments.  We revised manuscript based on your comments, and corrected parts were written by red fonts.

1.  The authors claimed that the probe has good selectivity for dopamine. I wonder that the Fe2+-dopamine complex forms strongly. The authors should discuss it. The interference of other metal ions such as Fe3+, Cu2+, Zn2+ and so on with fluorescence intensity of BDP-Fe2+ should be carefully evaluated. "The quenching efficiency was Fe2+ (6.5) > Ni2+ (25.0) > Co2+ (156.0) shown in Figure 3 (b)" How is the quenching efficiency calculated here? It should be clarified.

Answer: The phrase “quenching efficiency” was replaced by “fluorescence intensity” as shown in page 4, line 144.

2.  English editing was carried out using professional English editing service.

Reviewer 2 Report

In this paper, a new probe for targeting dopamine has been developed. In the absence of dopamine, the fluorescence intensity is low, while in the presence of dopamine, the fluorescence emission increases due to the reaction of Fe2+ with dopamine. I have several comments:

In page 1, line 33-35: However, dopamine is also known to cause various diseases related to the human brain and neurological disorders such as Parkinson’s disease, schizophrenia and HIV infection, which are increasingly affecting the general population and attracting considerable attention.

This sentence should be clarified. Dopamine is not a toxicant that causes many diseases but is related to a lot of diseases.

The authors claim that the sensitivity of the probe is very high. However, in the literature, there are also a lot of methods for dopamine determination with very high sensitivity. Please claim the novelty and specialty of this paper.

The authors claim that the fluorescence increment caused by dopamine was from free BDP. However, in fig.5, the fluorescence of BDP is not as high as dopamine+ BDP-Fe. Please explain this phenomenon.

In table 1, the authors should compare this probe with other probes.

Can you this probe use in biological samples? Will the Fe ion in biological sample interfere the detection?

Will pH affect the sensitivity and selectivity?

Author Response

Answer to Reviewer 2:

Thank you for your specific comments.  We revised manuscript based on your comments, and corrected parts were written by red fonts.

1. In page 1, line 33-35: However, dopamine is also known to cause various diseases related to the human brain and neurological disorders such as Parkinson’s disease, schizophrenia and HIV infection, which are increasingly affecting the general population and attracting considerable attention. This sentence should be clarified. Dopamine is not a toxicant that causes many diseases but is related to a lot of diseases.

Answer: To clarify this sentence, new sentence was written on page 1, line 31-36 as follows;Although dopamine itself is not a toxicant, it causes many diseases also causes various diseases related to the human brain and neurological disorders such as Parkinson’s disease, which is caused by the fact that Parkinson’s disease causes the reduction of dopaminergic neuron and dopamine is not made enough in human body. As a result, the movement of body cannot be adjusted and is impaired, which are increasingly affecting the general population and attracting considerable attention [5–8].

2.  The authors claim that the sensitivity of the probe is very high. However, in the literature, there are also a lot of methods for dopamine determination with very high sensitivity. Please claim the novelty and specialty of this paper.

Answer: In Table 1 and Table 2, this study was compared with our previous study and ELISA method.  From the view point of sensitivity and operation time, the dopamine analysis using BDP-Fe2+ improved, and satisfies the requirements for highly sensitive, highly selective, and high-throughput assays of dopamine.   

3.  The authors claim that the fluorescence increment caused by dopamine was from free BDP. However, in fig.5, the fluorescence of BDP is not as high as dopamine+ BDP-Fe. Please explain this phenomenon.

Answer: Y axis in new Figure 7 indicates that I: fluorescence intensity of BDP or BDP-M2+ at 525 nm before and after addition of dopamine, and I0: fluorescence intensity of BDP or BDP-M2+ themselves at 525 nm. In the case of BDP which is blank, fluorescence ratio = (fluorescence intensity of BDP) / (fluorescence intensity of BDP) = 1.0. In the same way, fluorescence ratio = (fluorescence intensity of BDP-Fe2+ after addition of dopamine) / (fluorescence intensity of BDP-Fe2+ itself) = 49.84 in the case of BDP-Fe2+ + Dopamine.  Therefore, the fluorescence ratio of BDP is not as high as that of BDP-Fe2+ + dopamine.

4.  In table 1, the authors should compare this probe with other probes. Can you this probe use in biological samples? Will the Fe ion in biological sample interfere the detection?

Answer: As an application of this assay, the detection of dopamine in human serum was conducted using BDP-Fe2+ and the results were compared with dopamine assays in HEPES buffer solution. Figure 11 shows the relationship between fluorescence intensities at 525 nm and dopamine concentration. The fluorescence intensity of BDP-Fe2+ in human serum or in the buffer solution increased with increasing dopamine concentration, and a good linear relationship was observed.  Thus, BDP-Fe2+ could be used to detect dopamine with high selectivity in the presence of other substances like BSA, inorganic salts, etc. and this assay is applicable to the detection of dopamine in the presence of impurities, and Fe2+ did not interfere the detection.  Above results were written on page 10, line 276-284.

5. Will pH affect the sensitivity and selectivity?

Answer: In order to understand how pH affect the reaction between BDP-Fe2+ and dopamine, fluorescence signals were monitored at different pH values. Figure 6 shows the fluorescence intensity at 525 nm of BDP-Fe2+ with 5.0 mM of dopamine. At high pH (10 to 11) and low pH (3 to4), the reaction of BDP-Fe2+ with dopamine was inhibited by the solution pH, whereas the fluorescence intensity was nearly unchanged when pH was in the range 5 to 9. As a result, the reaction of BDP-Fe2+ with dopamine was not inhibited nor masked at physiological level of pH values. This sentence was written on page 7, line 190-196.

6. English editing was carried out using professional English editing service.

Reviewer 3 Report

The manuscript describes a new fluorescent reagent for dopamine determination in serum. The synthesis of new chemosensors are relevant and the obtained results seems to be better than previous works. However, I have some considerations prior the publication. In my opinion, the manuscript should be accepted after major corrections.

Comments:

Line 82: Please specify which solvent was used.

Which volume of dopamine solution was added to the BDP-Fe2+ complex during calibration? How the limit of detection was calculated?

It is showed that Ni2+ and Co2+ have produced fluorescence quenching as well as Fe2+. How about other metals? Perhaps the quenching effect would be more effective with other metals or even more selective to dopamine?

Why the complex of BDP-Fe2+ is highly selective to dopamine? Despite of the reaction proposed in figure 6, it is not clear how dopamine bounds with the complex. The authors should include some discussions on this matter.

The authors mentioned previous compounds related to dopamine determination and claimed that BDP-Fe2+ has better performance among them. However, in my opinion the analytical performance comparison must be show as a table.

The time-dependence of BDP-Fe2+ / dopamine reaction is not evaluated. How long is the signal stable or how many time is needed to reach a stable signal?

Author Response

Answer to Reviewer 3:

Thank you for your specific comments.  We revised manuscript based on your comments, and corrected parts were written by red fonts.

1.  Line 82: Please specify which solvent was used.

Answer: The name of solvent was written on page 3, line 82 as follows: HEPES buffer solution (pH 7.0).

2.  Which volume of dopamine solution was added to the BDP-Fe2+ complex during calibration? How the limit of detection was calculated?

Answer: After the addition of 1.0 mL of dopamine or foreign substances solution to 1.0 mL of BDP-Fe2+ solution, fluorescence spectra were recorded during calibration. This method was written on page 3, line 83-85 as follows;

After the addition of 1.0 mL of dopamine or foreign substances solution to 1.0 mL of BDP-Fe2+ solution, the reaction mixture was incubated for 5 min, and fluorescence spectra were recorded.

The limit of detection was calculated as follows:

Concentration of dopamine: 500 nM

Fluorescence Intensity: 68.0

Baseline noise: 0.05

Limit of detection = 0.05×3×500(nM) / 68 = 1.1 (nM)

3.  It is showed that Ni2+ and Co2+ have produced fluorescence quenching as well as Fe2+. How about other metals? Perhaps the quenching effect would be more effective with other metals or even more selective to dopamine?

Answer: Fluorescence quenching effect and the reaction with dopamine were monitored after the additions of other transition metal ions (Cu2+, Mn2+, Zn2+), alkali metal ions (Li+, Na+, K+, Cs+), alkaline earth metal ions (Mg2+, Ca2+, Ba2+), Pb2+. In the case of Cu2+, Mn2+, Zn2+, fluorescence quenching was produced, whereas fluorescence enhancement was not observed after the addition of dopamine.  Moreover, both fluorescence quenching and fluorescence enhancement did not take place in the case of alkali metal ions (Li+, Na+, K+, Cs+), alkaline earth metal ions (Mg2+, Ca2+, Ba2+) and Pb2+.  As a result, Fe2+ played the important role for the detection of dopamine. Above results were shown in Figure 3 and 7, and were described on the text in page 4, line 142-147, and page 7 to line 197-209 as follows:

Other metal cations (Ni2+, Co2+, Cu2+, Mn2+, Zn2+, Li+, Na+, K+, Cs+, Mg2+, Ca2+, Ba2+, and Pb2+) were added to the solution of BDP, and the fluorescence intensities were monitored. The observed order of the fluorescence intensity was as follows: Fe2+ (6.5) > Ni2+ (25.0) > Co2+ (156.0) > Cu2+(201.5) > Mn2+ (250.2) > Zn2+ (292.3). Fluorescence quenching did not take place for alkali metal ions, alkaline earth metal ions, and Pb2+, as shown in Figure 3 (b). Thus, Fe2+ was a particularly strong quencher of BDP fluorescence.

To investigate the role of Fe2+ in BDP-Fe2+ for the detection of dopamine, the fluorescent properties of the free form of BDP (not Fe2+ complex), were monitored before and after the addition of dopamine (0 to 5.0 mM). Fluorescence intensities of BDP at 525 nm changed hardly after the addition of dopamine as shown in Figure 7. Moreover, other transition metal complexes (BDP-M2+ (M = Ni2+, Co2+, Cu2+, Mn2+, Zn2+)) did not show any responses after addition of dopamine because  no reaction occurred between  these metals and the dopamine [19]. On the other hand, BDP-Fe2+ indicated large fluorescence enhancement after the reaction with dopamine. From these results, Fe2+ in BDP-Fe2+ plays an important role in the detection of dopamine, and schematic representations of the reaction between BDP-Fe2+ dopamine were illustrated in Figure 8. The free form of BDP-Fe2+ emitted remarkably weak fluorescence, whereas the fluorescence enhancement by the interaction of BDP-Fe2+ with dopamine can be considered to be the release of Fe2+ from BDP-Fe2+, and the dopamine coordinated with Fe2+ in a 3:1 stoichiometry to form stable complex by the ligand exchange reaction [14, 19].

4.  Why the complex of BDP-Fe2+ is highly selective to dopamine? Despite of the reaction proposed in figure 6, it is not clear how dopamine bounds with the complex. The authors should include some discussions on this matter.

Answer: To investigate the complex formed between dopamine and the Fe2+ ion by a ligand exchange reaction, stoichiometry between dopamine and Fe2+ was investigated using absorption spectrophotometry in our previous report. As a result, a break point was observed at a mole fraction of 0.75 for dopamine in a Job’s plot, which indicates that dopamine coordinates with Fe2+ in a 3:1 stoichiometry to form stable complex. On the other hand, other metal ions did not react with dopamine.  Therefore, BDP-Fe2+ has high selectivity to dopamine. The above discussion was described on page 7, line 197-209 as follows:

To investigate the role of Fe2+ in BDP-Fe2+ for the detection of dopamine, the fluorescent properties of the free form of BDP (not Fe2+ complex), were monitored before and after the addition of dopamine (0 to 5.0 mM). Fluorescence intensities of BDP at 525 nm changed hardly after the addition of dopamine as shown in Figure 7. Moreover, other transition metal complexes (BDP-M2+ (M = Ni2+, Co2+, Cu2+, Mn2+, Zn2+)) did not show any responses after addition of dopamine because  no reaction occurred between  these metals and the dopamine [19]. On the other hand, BDP-Fe2+ indicated large fluorescence enhancement after the reaction with dopamine. From these results, Fe2+ in BDP-Fe2+ plays an important role in the detection of dopamine, and schematic representations of the reaction between BDP-Fe2+ dopamine were illustrated in Figure 8. The free form of BDP-Fe2+ emitted remarkably weak fluorescence, whereas the fluorescence enhancement by the interaction of BDP-Fe2+ with dopamine can be considered to be the release of Fe2+ from BDP-Fe2+, and the dopamine coordinated with Fe2+ in a 3:1 stoichiometry to form stable complex by the ligand exchange reaction [14, 19].

5.  The authors mentioned previous compounds related to dopamine determination and claimed that BDP-Fe2+ has better performance among them. However, in my opinion the analytical performance comparison must be show as a table.

Answer: Comparison of this study with previously reported compound were summarized on new Table 1, and was discussed on page 9, line 258 to page 10, line 266 as follows;

The analytical performance of BDP-Fe2+ was compared with those of previously reported compounds (References 1 and 2) and is summarized in Table 1. BDP-Fe2+ exhibited large fluorescence enhancement upon dopamine addition, whereas the fluorescence intensities of References 1 and 2 were approximately three times and ten times lower, respectively, than those of BDP-Fe2+. Thus, BDP-Fe2+ could be used to detect dopamine with greater sensitivity than References 1 and 2. These observations indicate that the BDP group in BDP-Fe2+ is a sensitive functional group, and contributes to larger fluorescence enhancement via ligand exchange between dopamine and BDP-Fe2+, and the chemical structure of the fluorescent dye in the fluorescent probe plays an important role in determining the fluorescent properties.

6.  The time-dependence of BDP-Fe2+ / dopamine reaction is not evaluated. How long is the signal stable or how many time is needed to reach a stable signal?

Answer: The time-dependence of the reaction between BDP-Fe2+ and dopamine was described on new Figure 5, and result and discussion were written on page 7, line 187-189 as follows:

Figure 5 shows the reaction time dependence of the fluorescence intensity at 525 nm of BDP-Fe2+ with dopamine. Fluorescence intensity saturated after 3.0 ~ 5.0 min, and 5.0 min was considered optimal for the reaction between BDP-Fe2+ and dopamine.

7.  English editing was carried out using professional English editing service.

Round 2

Reviewer 2 Report

The authors improved the manuscript a lot so I think the paper can be published as it as.

Author Response

Thank you for your specific comments.

English correction was underwent by MDPI English editing service.

Corrected parts were written by red fonts.

Reviewer 3 Report

The authors have corrected the manuscript and also replied all comments. However, I am still dubious concerning the limit of detection calculation. According to the author's reply: 

The limit of detection was calculated as follows:

Concentration of dopamine: 500 nM

Fluorescence Intensity: 68.0

Baseline noise: 0.05

Limit of detection = 0.05×3×500(nM) / 68 = 1.1 (nM)

Honestly I have never seen LOD being calculated like this before. According to the equation, LOD = 3xBN/S, where BN is the standard deviation of the signal of blank (baseline noise) and S is the slope of the analytical curve of the method. Please explain why this expression was not used. 

Author Response

Thank you for your specific comments.

The response to your comments is written as follows.

1.  The authors have corrected the manuscript and also replied all comments. However, I am still dubious concerning the limit of detection calculation. According to the author's reply:

The limit of detection was calculated as follows:

Concentration of dopamine: 500 nM

Fluorescence Intensity: 68.0

Baseline noise: 0.05

Limit of detection = 0.05×3×500(nM) / 68 = 1.1 (nM)

Honestly I have never seen LOD being calculated like this before. According to the equation, LOD = 3xBN/S, where BN is the standard deviation of the signal of blank (baseline noise) and S is the slope of the analytical curve of the method. Please explain why this expression was not used.

Answer: There are two methods to obtain LOD.  One is the calculation method using baseline noise, which I mentioned.  The other is the method using the standard deviation of the signal of blank and the slope of the analytical curve, which you mentioned above.  I calculated LOD by the standard deviation of the signal of blank and the slope of the calibration curve again, and almost the same numerical value (1.1 nM) was obtained.

2.      English correction was underwent by MDPI English editing service. Corrected parts were written by red fonts.